# Spatial distribution of epibionts on olive ridley sea turtles at Playa Ostional, Costa Rica

Nathan J. Robinson[1,2]*, Emily M. Lazo-Wasem[3], Brett O. Butler[4], Eric A. Lazo-Wasem[5], John D. Zardus[6], Theodora Pinou[7]

**1** The Leatherback Trust, Goldring-Gund Marine Biology Station, Playa Grande, Guanacaste, Costa Rica, **2** Cape Eleuthera Institute, The Cape Eleuthera Island School, Cape Eleuthera, Eleuthera, The Bahamas, **3** Lyles School of Civil Engineering, Purdue University, West Lafayette, Indiana, United States of America, **4** Museo de Zoología "Alfonso L. Herrera", Facultad de Ciencias, Universidad Nacional Autónoma de México, A.P. 70¬–399, Ciudad de México CP, México, **5** Division of Invertebrate Zoology, Peabody Museum of Natural History, Yale University, New Haven, Connecticut, United States of America, **6** Department of Biology, The Citadel, Charleston, South Carolina, United States of America, **7** Department of Biological and Environmental Sciences, Western Connecticut State University, Danbury, Connecticut, United States of America

* nathan@leatherback.org

**Data Availability Statement:** All relevant data are within the manuscript and its Supporting Information files.

## Abstract

There is a wealth of published information on the epibiont communities of sea turtles, yet many of these studies have exclusively sampled epibionts found only on the carapace. Considering that epibionts may be found on almost all body-surfaces and that it is highly plausible to expect different regions of the body to host distinct epibiont taxa, there is a need for quantitative information on the spatial variation of epibiont communities on turtles. To achieve this, we measured how total epibiont abundance and biomass on olive ridley turtles *Lepidochelys olivacea* varies among four body-areas of the hosts (n = 30). We showed that epibiont loads on olive ridleys are higher, both in terms of number and biomass, on the skin than they are on the carapace or plastron. This contrasts with previous findings for other hard-shelled sea turtles, where epibionts are usually more abundant on the carapace or plastron. Moreover, the arguably most ubiquitous epibiont taxon for other hard-shelled sea turtles, the barnacle *Chelonibia* spp., only occurred in relatively low numbers on olive ridleys and the barnacles *Stomatolepas elegans* and *Platylepas hexastylos* are far more abundant. We postulate that these differences between the epibiont communities of different sea turtle taxa could indicate that the carapaces of olive ridley turtles provide a more challenging substratum for epibionts than do the hard shells of other sea turtles. In addition, we conclude that it is important to conduct full body surveys when attempting to produce a holistic qualitative or quantitative characterization of the epibiont communities of sea turtles.

## Introduction

In the marine environment, almost any non-toxic, non-protected surface will eventually be colonized by an array of microorganisms, plants, algae, or animals [1,2]. This is true for non-living substrata, such as rocks, sand grains, or the shells of dead molluscs, as well as the bodies of living marine animals. Those organisms that live on the surfaces of other organisms are

**Funding:** The author(s) received no specific funding for this work.

**Competing interests:** The authors have declared that no competing interests exist.

referred to as epibionts [3], a grouping of diverse taxa with a wide range of life-histories, physiological tolerances, and mechanisms for attaching to their host [4]. It is therefore not surprising to discover that epibiont communities vary among different host species [5,6], populations [7], and even between different body-areas on a single host [8,9]. To understand why this variation occurs requires knowledge of the factors driving spatial variation in the distribution of epibionts, both on the scale of an ocean or a host's body.

Sea turtles arguably host the most abundant and diverse epibiont communities of all large marine megafauna. In an effort to characterize these epibiont communities, researchers have generated hundreds of species-lists reporting the presence of epibiont taxa for sea turtle populations around the world [10]. As these datasets have grown, there is now an impetus to synthesise these data and elucidate epibiont community spatial patterns globally and make quantitative assessments of epibiont loads [11]. Yet to help draw robust conclusions about geographic differences in epibiont communities, it would first be beneficial to understand spatial patterns in epibiosis at a finer-scale–the scale of a turtle's body. Indeed, it is highly likely that epibionts are non-uniformly distributed over a turtle host for several reasons. For example, the physical properties of certain body-parts, such as the carapace, differ from those of the plastron or skin [12], which likely affects attachment or settlement of different epibiont species. Different locations on the host's body may also present different opportunity for feeding. For example, some epibiont species feed preferentially on certain tissues and thus will be found predominantly in those areas [13] whereas filter feeders might prefer to be in areas where water flow over the host is optimal [14]. Either of these factors, as well as attachment mode, crypsis, mating habits, and others, could lead to distinct variation in the distribution of epibionts on sea turtles.

While it may be clear that epibiont distribution is likely to vary over the host's body, many studies on sea turtle epibiosis have either not provided any quantitative assessment of where on the host the epibionts were found (e.g.[7, 15]), and/or have simply sampled epibionts from a single location on the animal, commonly the carapace (e.g. [16,17]). Tellingly, even some of the most seminal studies investigating the distribution patterns of epibionts on the loggerhead turtle *Caretta caretta* and green turtle *Chelonia mydas* hosts have focused exclusively on the carapace [8,18]. There are only three studies of which the authors are aware that have compared the abundance of epibionts from over the entire body: Nájera-Hillman et al. (2012) [19], Devin and Sadeghi (2010) [20], Razaghian et al. (2019) [21]. Interestingly, these studies, which focused on either hawksbill turtles *Eretmochelys imbricata* or green turtles *Chelonia mydas*, identified that epibiotic barnacles were more common on the carapace and plastron than the soft tissues. If this is true for all sea turtle species, then this finding could have important implications for epibiont sampling. Specifically, if epibionts are predominantly found on certain body-parts of the host then then the actual biodiversity of epibionts might be underestimated if only sampled from a single region of the body.

To better understand spatial variation in epibiont communities on sea turtle hosts, we made a quantitative assessment of the abundance and biomass of several epibiotic taxa from different regions of the body of olive ridley sea turtles *Lepidochelys olivacea*. This species was chosen because several studies have focused on characterizing the epibiont communities of olive ridleys in recent years [6,15,22,23] yet no previous studies have specifically looked at epibiont distributions on this species. Our null prediction was that there would be no variation in the abundance or total biomass of the various epibiont taxa among the different body-areas.

## Methods

### Study site

Olive ridley turtles were sampled from Playa Ostional, which is located on the Pacific coast of the Nicoya Peninsula in northwest Costa Rica (9° 59' N, 85° 42' W). We selected this site as it

is one of a few beaches worldwide that has mass-nesting assemblages or "*arribadas*" of olive ridley turtles, thus making it feasible to sample large numbers of turtles relatively easily. These arribadas generally occur once a month and last three to eight days [24]. At Ostional, over 500,000 turtles have been estimated to nest in a single arribada event [25] at densities of over 4 clutches m$^{-2}$ [26]. For this study, we sampled epibionts from a total of 30 turtles spanning three separate arribada events. Specifically, we sampled 10 turtles on 6-Dec-2015, 5 turtles on 1-Jan-2016, and 15 turtles on 7-Feburary-2016.

## Sampling protocol

During arribada events, we patrolled the beaches of Ostional to encounter nesting turtles. To avoid any possibility that sampling would interrupt the nesting process, we only examined turtles after they had completed oviposition. In addition, we sampled only animals that appeared to be in good health with no visible injuries, lesions, or tumours as certain epibionts are often found in association with certain injuries [27,28].

When a suitable turtle was encountered, we measured its Curved Carapace Length (CCL) using a flexible tape measure. The turtle's flippers were then restrained by hand and the animal was moved onto a plastic tarp. This limited the ability of the turtle to flick sand on its carapace, which would impede efforts to sample epibionts. We then exhaustively collected epibionts by scraping or prying them off the turtle using a knife or tweezers. After all visible epibionts were collected from the dorsal surfaces of the turtle, we would briefly (< 10 mins) flip the animal onto its carapace to collect epibionts from its ventral surfaces. All epibionts were divided between separate containers by the region of the body where they were collected. We divided the body into four regions: (1) the head, shoulder, and fore flippers (subsequently termed head), (2) the tail, cloaca, hind flippers, and inguinal cavities (subsequently termed tail), (3) the carapace, or (4) the plastron (Fig 1). Epibiont samples were preserved in 75% non-denatured ethanol following the protocols outlined in Lazo-Wasem et al. (2011) [15]. Due to the high numbers of turtles at each arribada, we did not deem it necessary to tag individual turtles. As we only conducted one night of sampling per arribada, it was easy to ensure that we were not sampling the same individual multiple times by consistently only sampling individuals as we moved in one direction down the beach. Moreover, as up to 500,000 individuals have been recorded during a single arribada at Playa Ostional [25], the probably that we re-sampled an individual that had already been encountered on a previous arribada was negligible.

All epibionts were identified to the lowest taxonomic level by consulting appropriate literature. The samples were then enumerated, dried, and weighed to the nearest 0.00001g using a Mettler B6 Semi-Micro Balance. Samples were catalogued and deposited in the Yale Peabody Museum of Natural History, U.S.A. Specimen records are available at http://peabody.yale.edu/collections.

To test our null prediction that there would be no variation in the abundance or total biomass of the epibiont taxa between the different regions of the turtles' body, we used a one-way PerMANOVA using the Bray–Curtis index of dissimilarity. We choose a PerMANOVA test over other methods for examining difference in community structure, such as ANOSIM or Mantel tests, because PerMANOVA tests are considered more robust to difference in multivariate dispersion [29]. Significance was computed through permutation by group membership with 9999 replicates and applying the Bonferroni correction to account for biases associated with multiple comparisons.

After using the one-way PerMANOVA to identify statistical differences in the abundance and biomass of the epibiont community between different body-parts, we used pair-wise Mann-Whitney tests to assess specifically how the different epibiont taxa vary in abundance

**DORSAL VIEW**  **VENTRAL VIEW**

**Fig 1. An illustration of the four different regions of the body that were sampled for epibionts.** (1) The head, shoulder, and fore flippers (= **head**), (2) The tail, cloaca, hind flippers, and inguinal cavities (= **tail**), (3) The shell dorsum (= **carapace**), and (4) The shell ventrum (= **plastron**),.

and biomass between the different body-parts. Mann-Whitney tests were conducted using PAST V.3.13, confirming significance when $p \leq 0.05$. We chose a non-parametric Mann-Whitney test, over its parametric counterparts, because the data included many zeros and were not normally distributed.

## Results

Of the 30 olive ridley turtles that were sampled (CCL: 66–69 cm range), epibionts were found on all but one individual. In total, we collected 1614 individual epibionts, which collectively weighed 14.01 g. The mean number of epibionts per host was 53.80 with a mean cumulative mass of 0.47 g. The epibiont load by body region (considering all epibiont taxa) was highest on the tail with 59.23% of epibionts being found in this location and constituting 64.11% of the total epibiont biomass (Fig 2). The location with the next highest epibiont load was the head with 31.69% of epibionts being found in this location and constituting 24.83% of the total epibiont biomass. For both the carapace and plastron, epibiont load was very low and never exceeded more than 8% of the total epibiont load.

We conservatively estimate that the diversity of epibionts represented 20 different taxa (see Table 1 for a full list). Most of these taxa were relatively rare and only five were found in mean abundances exceeding one individual per host. In alphabetical order these were: *Balaenophilus manatorum*, *Chelonibia testudinaria*, *Platylepas hexastylos*, *Stomatolepas elegans*, and small individuals of family Platylepadidae that cannot as yet be identified. This platylepadid may comprise more than one species, but we took the most conservative approach of viewing it as a single operational taxonomic unit.

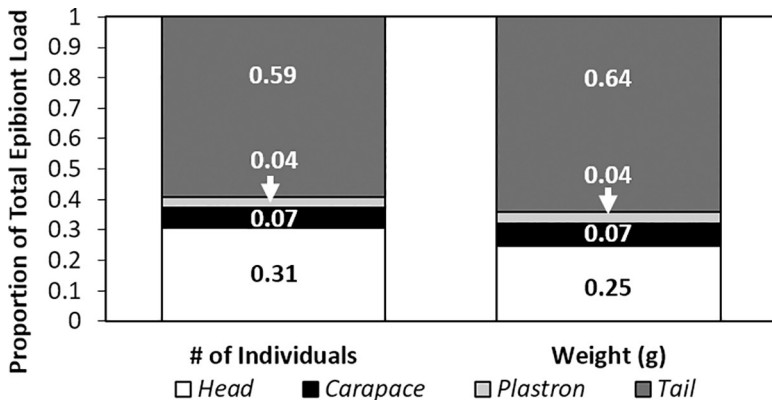

**Fig 2. Proportion of the total epibiont load found on the four body regions (head, carapace, plastron, and tail) of olive ridley sea turtles.** The left graph represents % abundance and the right graph represents % total biomass.

The most common taxon was *S. elegans*, which constituted 38.85% and 63.30% of the total epibiont load in terms of abundance and mass respectively. *Platylepas hexastylos* was almost as abundant, constituting 29.30% of all epibionts but individuals were generally lower in mass than *S. elegans* and so constituted only 17.24% of the total epibiont biomass. In contrast, *C. testudinaria* was not very abundant, comprising only 5.27% of all epibionts, yet it constituted 16.54% of total epibiont biomass. *B. manatorum* was relatively abundant, constituting 17.66% of all epibiont individuals, yet the combined biomass of this species was less than 0.01 g and so relatively negligible. Similarly, Platylepadidae spp. *c*onstituted 5.70% of the total epibiont abundance yet only 0.8% of the total epibiont biomass, further indicating that these individuals were very small. The remaining 15 taxa constituted only 3.34% and 2.09% of the total epibiont load in terms of abundance and biomass respectively (Fig 3).

The PerMANOVA test identified a statistically significant difference in the epibionts communities between the four-body parts, in terms of both abundance (F = 4.506, p < 0.001) and biomass (F = 3.598, p < 0.001). Pair-wise comparisons using the Bonferroni correction identified that these differences were not significant between the head and tail (abundance: p = 0.386; biomass: p = 0.331) nor between the carapace and plastron (abundance: p = 0.434; biomass: p = 0.850). However, significant differences were observed between the head or tail when compared to either the carapaces or plastron (all tests p < 0.002). Thus, we can reject the null prediction that epibionts are found in equal abundances on the different body-parts of the host (Fig 4).

When we tested for differences in the abundance or biomass of the five major epibiont taxa between the different body region, we observed significant differences in the abundance and biomass of *S. elegans* (abundance: $H_{(\chi 2)}$ = 25.49, d.f. = 3, p < 0.0001; biomass = $H_{(\chi 2)}$ = 15.67, d.f. = 3, p < 0.0001) and *P. hexastylos* (abundance: $H_{(\chi 2)}$ = 16.87, d.f. = 3, p < 0.0001; biomass = $H_{(\chi 2)}$ = 15.67, d.f. = 3, p < 0.0001). Ad-hoc pair-wise Mann-Whitney tests revealed that for both *S. elegans* and *P. hexastylos* there were no statistical differences in either abundance or biomass between head or tail sections (p < 0.0001 for all tests). Instead, statistical differences were observed in abundance or biomass between both the head and tail when compared to either carapace or plastron (p > 0.5 for all tests).

No significant differences were observed in the abundance or biomass of *C. testudinaria* (abundance: $H_{(\chi 2)}$ = 2.94, d.f. = 3, p = 0.1973; biomass = $H_{(\chi 2)}$ = 1.613, d.f. = 3, p = 0.4650), *B. manatorum* (abundance: $H_{(\chi 2)}$ = 0.2748, d.f. = 3, p = 0.5139; biomass = $H_{(\chi 2)}$ = 0.2727, d.f. = 3, p = 0.5173), and Platylepadidae spp. (abundance: $H_{(\chi 2)}$ = 0.4883, d.f. = 3, p = 0.5830; biomass = $H_{(\chi 2)}$ = 0.4834, d.f. = 3, p = 0.5873) between the different body areas.

**Table 1. Mean # and biomass of the different epibiont species found on 30 olive ridley sea turtles sampled from Playa Ostional, Costa Rica.**

| Systematic Group | Epibiont Taxon | Mean # of epibionts per host | Mean epibiont biomass per host (g) |
|---|---|---|---|
| **Annelida: Hirudinea** | *Ozobranchus branchiatus* | 0.1 | < 0.1 |
| **Annelida: Polychaeta** | Polychaeta | < 0.1 | < 0.1 |
| **Annelida: Polychaeta** | Serpulidae | 0.1 | < 0.1 |
| **Chlorophyta** | Algae / Hydroid | 0.3 | 0.1 |
| **Chelicerata: Pycnogonida** | Pycnogonida | < 0.1 | < 0.1 |
| **Crustacea: Amphipoda** | Caprellidea | 0.1 | < 0.1 |
| **Crustacea: Amphipoda** | Corophiidae | < 0.1 | < 0.1 |
| **Crustacea: Amphipoda** | *Podocerus chelonophilus* | < 0.1 | < 0.1 |
| **Crustacea: Brachyura** | *Planes* spp. | < 0.1 | < 0.1 |
| **Crustacea: Cirripedia** | *Chelonibia testudinaria* | 2.8 | 2.3 |
| **Crustacea: Cirripedia** | Chelonibiidae | < 0.1 | < 0.1 |
| **Crustacea: Cirripedia** | *Conchoderma virgatum* | < 0.1 | < 0.1 |
| **Crustacea: Isopoda** | Corallanidae | < 0.1 | 0.1 |
| **Crustacea: Cirripedia** | *Balanoidea* sp. | 0.5 | < 0.1 |
| **Crustacea: Cirripedia** | Platylepadidae spp. | 2.1 | 0.1 |
| **Crustacea: Cirripedia** | *Platylepas hexastylos* | 15.7 | 2.4 |
| **Crustacea: Cirripedia** | *Platylepas* sp. | 0.2 | < 0.1 |
| **Crustacea: Cirripedia** | *Stomatolepas elegans* | 20.9 | 8.9 |
| **Crustacea: Copepoda** | *Balaenophilus manatorum* | 8.7 | < 0.1 |
| **Crustacea: Copepoda** | Harpacticoida | < 0.1 | < 0.1 |
| **Mollusca: Gastropoda** | Unknown sp. | < 0.1 | < 0.1 |

## Discussion

### Spatial distribution of epibionts

Knowledge of how sea turtle epibionts are distributed on their hosts can provide valuable insights into the ecology and habitat requirements of these extracorporeal companions. To

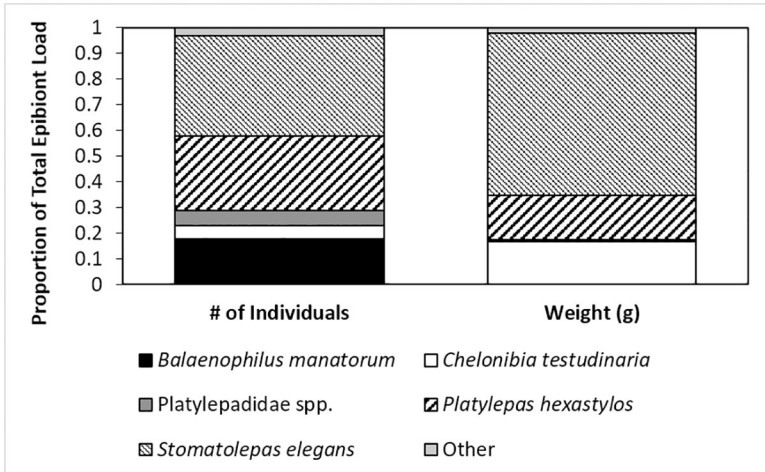

**Fig 3. Proportion of the five main epibiont species relative to the total epibiont load found on the entire body of olive ridley sea turtles.** The left graph represents % abundance and the right graph represents % total biomass.

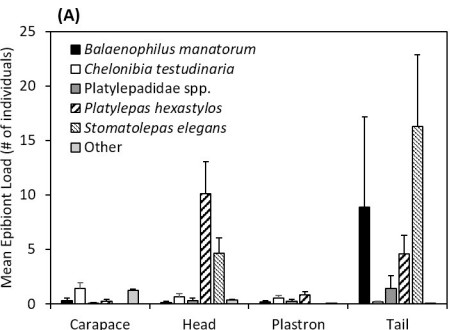
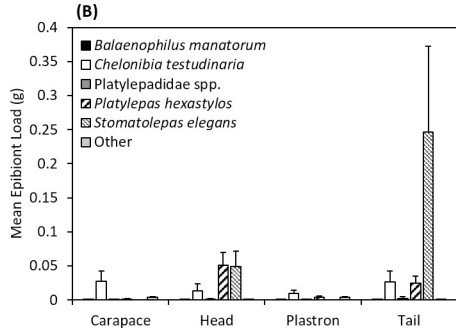

**Fig 4.** Mean number of individuals (A) and mean biomass (B) for the five main epibiont species (*Balaenophilus manatorum*, *Chelonibia testudinaria*, Platylepadidae spp., *Platylepas hexastylos*, and *Stomatolepas elegans*) from the four 4 different body regions (head, carapace, plastron, and tail) of olive ridley sea turtles.

investigate the spatial distribution of epibionts on olive ridley turtle hosts, we measured and compared the abundance and biomass of epibionts on different body regions. We discovered that there is indeed a statistically significant difference in the epibiont communities on the different body-regions of olive ridley sea turtles, specifically between the body-parts where the substrate is made of skin (i.e. the head and tail) and those of the shell (i.e. carapace and plastron). We also discovered that, in contrast to all other sea turtles species that have been assessed previously, epibiont loads on olive ridley turtles are highest, both in terms of number and biomass, on the soft tissues of the head and tail than they are on the carapace or plastron. Specifically, 60% of all epibionts were found on the tail and 32% were found on the head. In contrast, only 6% of all epibionts were found on the carapace and 4% on the plastron.

This finding suggests that for studies of sea turtle epibionts if the whole body is not sampled it can run the risk of under-sampling and misrepresenting certain epibiont taxa, especially in quantitative evaluations. For example, the most common epibiont taxon recorded in this study in terms of both abundance and biomass, *S. elegans*, was absent from the carapace and plastron. Similarly, and somewhat predictably due to the intimate association between *B. manatorum* and *S. elagans* [26], *B. manatorum* was largely absent from the carapace and plaston and only reported in substantial numbers on the tail section. Consequently, if our sampling strategy was exclusively focused on the turtles' shell then these two species would have been overlooked almost entirely. In the case of *B. manatorum*, this could be particularly important as *B. manatorum* has been previously suggested as a vector for fibropapillomatosis [27] and this prediction has been further supported by more recent studies [13,28]. Information on the presence or absence of this *B. manatorum* from different turtle populations could therefore help to reveal important insights into both the spread of this disease and to identify which populations are at the highest risk. This highlights the value of sampling the whole individual, and not only focus on specific body-parts, to achieve a holistic understanding sea turtles' epibiont communities.

We suspect that the primary reason that epibiont load and community structure vary so distinctly between the head or tail and the carapace or plastron is due to the differing surface properties of these body-parts. Indeed, among many differences, the 'soft-tissue' skin of the head and tail likely provide a very different surface for settlement than the keratinized scutes of the carapace. This factor could easily explain variation in the abundance of several key species, such as *S. elegans* and *C. testudinaria*. *Stomatolepas elegans* specifically was found in abundance on the head and tail but was absent on the carapace and plastron. In every instance, *S. elegans* was found embedded into the skin, where it was held in place by an array of plate-like

protuberances (for microscope imagery see [15]). While such an attachment mechanism is suited for embedding into soft substrates, such as the skin of the inguinal cavity, it is likely that such 'embedding' is not possible in the carapace or plastron. In contrast, among coronuloid barnacles that are obligate with sea turtles (see [5]), *C. testudinaria* is uniquely adapted solely for gluing to the carapace or plastron of its host by 'cementing' its membranous base to the substratum [30]. It does not penetrate or embed in any way as do other coronuloids, especially those that live in the skin, explaining why it was found most commonly on the carapace.

Another factor influencing the spatial distribution of the epibionts observed in this study could be the interactions among epibiota. As mentioned earlier, *B. manatorum* appear to be strongly associated with *S. elegans* and may even parasitize or feed directly on these barnacles [27], and thus we may expect to find *B. manatorum* in areas where *S. elegans* is abundant. There may also be several interactions between many of the macro-epibionts, our focused in this study, and other micro-epibionts that we did not assess here. Indeed, it has recently been discovered that sea turtles host a diverse array of epibiotic diatoms and bacteria [23,31]. As these species can form biofilms that may alter the surface properties of the substratum, which in turn can influence the settlement of macro-epibiota [32], it is possible that spatial variation in the hosts' biofilm could affect spatial distributions of the larger epibionts. While no studies have yet quantitatively assessed spatial variation in epibiotic diatoms or bacteria in sea turtles, we believe this would be a productive avenue for future research.

## Differences between host species

The discovery that epibiont loads are higher for olive ridley turtles on the soft tissues, such as the head and tail, relative to the carapace and plastron was somewhat unexpected. Indeed, comparable studies that conducted quantitative assessments of epibiont distributions on both green and hawksbill turtles found the majority of epibiotic barnacles on the carapace and plastron [19,20,21]. Furthermore, it has often been assumed although there is little quantitative data to support it, that for loggerhead turtles the highest quantities of epibionts are found on the carapace (see [33]).

This suggests that olive ridley carapaces might pose a more challenging substratum for epibiont settlement than that of other sea turtle species. This hypothesis is supported by our observation of the paucity of *Chelonibia* barnacles associated with this olive ridley turtle. *Chelonibia* spp. are common taxa on the carapaces of loggerhead turtles [16, 17], hawksbill turtles *Eretmochelys imbricata* [34], and green turtles [6]; however, we observed a mean of only 2.8 *C. testudinaria* per host, far lower than the observed 16.8 individuals per host on green turtles sharing nesting beaches with olive ridley turtles in Costa Rica [6]. The *C. testudinaria* found on olive ridley turtles were also far smaller than those on these green turtles (NJ Robinson, personal observation) suggesting that maybe they are not surviving long enough on this host to grow to full size.

These differences in epibiont settlement patterns could be driven by habitat preference and behaviour of each host species, exposing themselves to settlement by larval epibionts from different habitats (i.e. reef, seagrass meadow, or soft sediment), yet we consider this to be unlikely considering the large range and behaviour overlaps between many turtle species. Instead, we propose that differences in the surface properties (e.g. microstructure, chemical composition, physical properties, or biofilms) of the carapace and dermis among different turtle species could be driving spatial variation in epibiont settlement, confirmation of which would require further studies.

## Geographic patterns in epibiont communities in olive ridleys

In recent years, the epibiont communities of olive ridley sea turtles in the East Pacific Ocean have been described in several studies [6,15,22,23]. Collectively between these four studies, a

total of 291 olive ridley turtles have been sampled for epibionts. From this extensive dataset, it is clear that while the epibiont communities of olive ridley turtles in the East Pacific are diverse and that new epibiont associations have been revealed, these epibiont communities are generally dominated by a few main taxa. They include: the barnacles *C. testudinaria*, *C. virgatum*, *Lepas* spp. *P. decorata* (which we describe as Platylepadidae spp. in this study for reasons described earlier), *P. hexastylos*, and *S. elegans*; the leech *O. branchiatus*, the amphipod *P. chelonophilus*, and the copepod *B. manatorum*. Each of these taxa are obligate epibionts of marine vertebrates. All other taxa tend to be recorded at low frequencies (i.e. less than one per 50 hosts).

This is not to say, however, that more commonly observed taxa for East Pacific olive ridley turtle are found on every turtle. In fact, even these commonly observed epibiont taxa are noticeably absent from some studies. For example, *P. chelonophilus* was observed on more than 25% of the olive ridley turtles sampled both on the beach of Playa Grande [6] and in the nearby waters El Coco, Costa Rica [22]. However, in the present study on Playa Ostional, we recorded only a single individual of *P. chelonophilus* among all 30 turtles, even though this beach is less than 50 km away from the previously mentioned sampling locations. Majewska et al. (2015) also reported a similarly low prevalence of *P. chelonophilus* from olive ridley turtles sampled at Playa Ostional, with only 4% of all the turtles hosting this epibiont [23]. *Lepas* spp. are also noticeably less prevalent at Playa Ostional, relative to nearby locations [6,22,23].

One possible hypothesis to explain differences in the abundances of these common epibiont taxa between turtles sampled on Playa Ostional compared to those sampled at other locations nearby is that Playa Ostional if affected by different oceanographic conditions. While previous studies have indeed noted the presence of strong offshore currents on Playa Ostional, relative to other nesting beaches on the Pacific coast of Costa Rica [35], it seems unlikely that this is the primary factor driving the differences in epibiont communities between these locations. This is because nesting olive ridley turtles are known to move over larger distances between nesting events than the distances between these beaches (< 50 km) [36], and thus there is likely significant overlap in the inter-nesting habitats of these turtles even though they nest on different beaches. Instead, we propose the more robust hypothesis is that the differential prevalence of certain epibiont taxa is linked to differences in behaviour between turtles nesting at each site. Indeed, Playa Ostional is a well-known arribada site where thousands of turtles aggregate in mass nesting events, while Playa Grande and the waters of El Coco likely host turtles that are solitary nesters. Perhaps during such mass mating events there is a much greater chance for epibionts to be scraped off than there is in smaller mating events, meaning the arribada nesting turtles have fewer epibionts. Alternatively, as many epibionts, such as *P. chelonophilus* and *B. manatorum* are not free swimming and must likely be transferred on contact between hosts [28], it could be that the great potential for the transfer of individuals between hosts means that they even though they might be found on more hosts, they occur in lower numbers per host. Understanding the factors that are driving distinct variation in the presence and absence of these common epibiont species presents an interesting avenue for future research that could reveal insights about the behaviour of their hosts that might be undetectable using more conventional methods [28,37].

## Acknowledgments

Epibiont collection was conducted under MINAE permits (ACT-ORDR-099-15 and ACG-PI-058-2015). This research was performed in accordance with the Purdue University Animal Care and Use Committee. Samples were exported from Costa Rica under a SINAC permit

(DGVS-183-2016). Pilar Santidrián Tomillo, Christian Diaz Chuquisengo, Elizabeth Solano, and Myriam Norori helped with permitting. Lourdes Rojas aided with identifying taxa.

## Author Contributions

**Conceptualization:** Nathan J. Robinson, Eric A. Lazo-Wasem, John D. Zardus, Theodora Pinou.

**Formal analysis:** Nathan J. Robinson.

**Investigation:** Nathan J. Robinson.

**Methodology:** Nathan J. Robinson, Emily M. Lazo-Wasem, Brett O. Butler, Eric A. Lazo-Wasem.

**Writing – original draft:** Nathan J. Robinson.

**Writing – review & editing:** Eric A. Lazo-Wasem, John D. Zardus, Theodora Pinou.

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
