## [Decision Letter · Decision Letter 0]

2 Jul 2019

PONE-D-19-15745

Spatial distribution of epibionts on olive ridley sea turtles at Playa Ostional, Costa Rica

PLOS ONE

Dear Dr. Robinson,

Thank you for submitting your manuscript to PLOS ONE. After careful consideration, we feel that it has merit but does not fully meet PLOS ONE’s publication criteria as it currently stands. Therefore, we invite you to submit a revised version of the manuscript that addresses the points raised during the review process.

We would appreciate receiving your revised manuscript by Aug 16 2019 11:59PM. To enhance the reproducibility of your results, we recommend that if applicable you deposit your laboratory protocols in protocols.io, where a protocol can be assigned its own identifier (DOI) such that it can be cited independently in the future. For instructions see: http://journals.plos.org/plosone/s/submission-guidelines#loc-laboratory-protocols

We look forward to receiving your revised manuscript.

Kind regards,

Benny K.K. Chan, Ph.D

Academic Editor

PLOS ONE

Journal Requirements:

Additional Editor Comments:

Dear authors,

Your MS will need revisions. Please revise the MS and send back a letter point by point how the comments were addressed. I will send to the same reviewers for comments again.

Best wishes,

Benny

Reviewers' comments:

Reviewer's Responses to Questions

**Comments to the Author**

1. Is the manuscript technically sound, and do the data support the conclusions?

Reviewer #1: No

Reviewer #2: Yes

2. Has the statistical analysis been performed appropriately and rigorously? 

Reviewer #1: No

Reviewer #2: Yes

3. Have the authors made all data underlying the findings in their manuscript fully available?

Reviewer #1: Yes

Reviewer #2: Yes

4. Is the manuscript presented in an intelligible fashion and written in standard English?

Reviewer #1: Yes

Reviewer #2: Yes

5. Review Comments to the Author

Reviewer #1: The manuscript explores epibiont distribution on olive ridley sea turtles. The study is interesting but there are some critical problems to be corrected.

The manuscript claims that authors made quantitative analyses on the epibiont communities. However, the number of individuals and biomass should be divided by areas (cm2) of each body part, i.e., head, carapace, plastron, and tail, for the fair comparison.

It is unclear that how the p-values were corrected for the multiple comparison.

The manuscript lacks the discussion on the factors driving spatial variation in the distribution of epibionts.

I recommend to conduct more analyses on epibiont community structures, such as comparison of diversity index, and community compositions with nMDS ordination plot.

Reviewer #2: Review, Robinson et al.

Dear editor,

This manuscript provides results that suggest that sampling of epibionts of sea turtles should not only take place from the carapace or plastron, but indeed from all areas of the turtle, including the softer parts. Although the majority of epibionts confined to barnacles, the data (and the methods) might be useful for biologists working with conservation and epizoic organisms in general. I liked the approach and it is merited that we need to consider epibiont communities on the whole host rather than just the most convenient area. I think the authors write quite well around the problem of “sampling bias” if certain structures are only sampled.

I have some comments but recommend publication after a minor revision. The language is sound and sufficiently scientific, the perspective for the readers of Plos One is wide enough and the tests and field experiments seems appropriate.

Comments:

It would be nice with some photo material of the turtles and their epibionts. One of the advantages of publishing in Plos is that they provide high quality photo plates. Not many people might know what barnacles have been found and this might help other conservation biologists with limited time/lust for taxonomic identification of epibionts.

L49: not wide-range, but wide range

L53: insert can shed light “on”..

L54: What is varied associates really? Improve language

L55: How can the epibionts tell anything about the behavior of the hosts? Explain

L129-130: How can you really know? I think this argument is kind of vague. Think it is better to re-formulate or further explain how this “ensurance” is established. As I understand it, the area within each arribada where sampling took place is quite “small”

L145: did the authors define CCL upon first mentioning? I can’t find it, so please explain first and then abbreviate

L163 and onwards: Write genus name as single letter after first mentioning - this is proper taxonomic standard

L170: Why do you call it Platylepadidae? Is it referring to a familiy name (-dae)? Then why italics? I do not know the code of the zoological nomenclature entirely, but this seems odd.

L175: Test tests?

L222-225: What is this “ecology” the authors refer to here? What is “ecological importance”? I find this very vague and it doesn’t really say anything substantial

L227-228: Yes, the authors and right and they proved their point. However, I would like to see a discussion of potential epibionts that could have been overlooked. What about parasites embedded within the skin? What about “microorganisms” as the authors write about in the introduction? I do not know, but is there any knowledge on the bacteria on sea turtles? Can these be studied with the same approach here and species determined? This part could be relatively short and precise, but it might be good to consider that this method does not grant access to all “epibionts” per se

L241: I do not understand this expectation. Sea turtles (perhaps not this species?) shed their scutes and some even their skin. Any epibiont, particularly permanently attached barnacles would then fall off during successive molts. A natural expectation would be that the scutes at anytime hold fewer epibionts given that it is regularly replaced. I think the authors should write in the introduction that sampling only on the parts that are actually being shed might result in significant “sampling bias”. But I can appreciate if the authors are careful with such statements.

L255: it would be nice with a reference here and some elaboration of how turtle species differ in these properties. It appears a bit vague.

L263: The authors might have a point, but it cannot be out ruled that they spotted juvenile Chelonibia. I would mention this.

L266-267: This needs a reference or proper photo documentation. Is it the cyprid larva that digs into the tissue or is it the adults? Is the burrowing similar to the balanomorphan barnacle Xenobalanus globicipitis that settles and burrows into the integument of whales? If no, what is the difference? Have the two modes of “digging” been observed?

L306: This is plausible, but as the authors note, barnacles were scraped off with instruments. Barnacles generally attach firmly. I do not think it is likely that they are being scraped off, although some might. Would selection not impede settlement in those areas if it affected the survival of the species (i.e., they fall off and die)? Just a thought. I can buy it if the authors do not wish to go into detail.

L420: This reference has a bracket around it.

I recommend publication after a minor revision.

6. PLOS authors have the option to publish the peer review history of their article (what does this mean?). If published, this will include your full peer review and any attached files.

Reviewer #1: No

Reviewer #2: No

---

## [Author Response · Author response to Decision Letter 0]

24 Jul 2019

Dear Editor,

Thank you for reviewing the manuscript entitled “Spatial distribution of epibionts on olive ridley sea turtles at Playa Ostional, Costa Rica” that my colleagues and I submitted for publication in PLoS ONE. We are grateful for the insightful feedback of the two reviewers – their input has helped to substantially improved this manuscript.

Following the comments of the reviewers, we made two major changes to the manuscript. Firstly, we incorporated a PerMANOVA test to statistically demonstrate that differences in the epibiont community structure between the different body-regions. We included this test largely in response to the comments from Reviewer 1 that “I recommend to conduct more analyses on epibiont community structures, such as comparison of diversity index, and community compositions with nMDS ordination plot.”. We chose this test instead of adding diversity indices or nMDS plots as we believe that it better answers the overarching goal of this study, which was to determine whether selectively sampling for epibionts on a specific body-region of a sea turtles is a suitable method for gaining a holistic understanding of the individuals entire epibiota. Secondly, we honestly agreed with Reviewer 1’s comment that “The manuscript lacks the discussion on the factors driving spatial variation in the distribution of epibionts.” We are embarrassed by this oversite as this was indeed in the title of the manuscript and while it was discussed in the introduction, this discussion was not suitably continued into the Discussion. Thus, we have re-written much of the Discussion to provide more detailed, and hypothesis-driven, discussion on the factors that drive spatial variation in the distribution of epibionts on sea turtles. 

Below, we will provide a point-by-point response as to how we incorporated all of the other feedback provided by the reviewers:

Reviewer 1

We follow the logic of Reviewer 1’s first comment, “The manuscript claims that authors made quantitative analyses on the epibiont communities. However, the number of individuals and biomass should be divided by areas (cm2) of each body part, i.e., head, carapace, plastron, and tail, for the fair comparison.”. However, we disagree that this is the appropriate course of action in this manuscript. The goal of this manuscript was to determine selectively sampling for epibionts on a specific body-region of a sea turtles is a suitable method for gaining a holistic understanding of the individuals entire epibiota. Thus, it is more appropriate to assess total epibiont load, rather than epibiont density, on the different body-parts. Similar methods have also been used in all several other studies on epibiont distribution (e.g. Nájera-Hillman et al. 2012, Devin and Sadeghi 2010, Razaghian et al. 2019). Finally, we cannot think of a practical method for estimating the entire surface of a turtle in the field and would be very interested in hearing the reviewers thought on how to achieve this. It may be possible to generate a very rough estimate based on the Curved Carapace Length of the sampled individuals and then applying this to a digital 3D models of an olive ridley sea turtle (see Digital Life 3D for examples); however in doing so, we believe that we would include so much additional, and unnecessary, error that would likely generate very inaccurate results.

We were a little unsure as to how to address the following comment “It is unclear that how the p-values were corrected for the multiple comparison.” as each test was an independent test and thus there was no need to correct for multiple comparisons. Nevertheless, we do agree that a multivariate test to statistical determine whether the epibiont communities differed between body-regions was necessary and so included the PerMANOVA test mentioned earlier. 

Overall, we strongly appreciate the criticisms of Reviewer 1 as it has helped us to address some of the weaker parts of this study. We also encourage Reviewer 1 to provide more details in their subsequent reviews that will help us to make the necessary edits that will ensure that science in this manuscript is of the highest quality possible.

Reviewer 2

We incorporated all over Reviewer 2’s grammatical and typographical edits into the text. Thus, we will only discuss those comment that require more substantial edits.

• “L55: How can the epibionts tell anything about the behavior of the hosts? Explain” One example would be the animals with different habitat preferences may have different epibionts. For example, turtles that live in the offshore habitats may have different epibiont to individuals in coastal habitats. Thus, the epibiota of nesting sea turtles can indicate where they live. Nonetheless, we actually removed the statement from the text as we did not believe that is was necessary to make this link in the first paragraph of the Introduction.

• “L129-130: How can you really know? I think this argument is kind of vague. Think it is better to re-formulate or further explain how this “ensurance” is established. As I understand it, the area within each arribada where sampling took place is quite “small”” We have addressed this issue in the text.

• “L222-225: What is this “ecology” the authors refer to here? What is “ecological importance”? I find this very vague and it doesn’t really say anything substantial” We agree that these statements do not say anything substantial and so have removed them from the Discussion.

• “L227-228: Yes, the authors and right and they proved their point. However, I would like to see a discussion of potential epibionts that could have been overlooked. What about parasites embedded within the skin? What about “microorganisms” as the authors write about in the introduction? I do not know, but is there any knowledge on the bacteria on sea turtles? Can these be studied with the same approach here and species determined? This part could be relatively short and precise, but it might be good to consider that this method does not grant access to all “epibionts” per se” – We have now included more discussion with regards to micro-epibionts in the Discussion.

• “L241: I do not understand this expectation. Sea turtles (perhaps not this species?) shed their scutes and some even their skin. Any epibiont, particularly permanently attached barnacles would then fall off during successive molts. A natural expectation would be that the scutes at anytime hold fewer epibionts given that it is regularly replaced. I think the authors should write in the introduction that sampling only on the parts that are actually being shed might result in significant “sampling bias”. But I can appreciate if the authors are careful with such statements.” We do not know of any literature on the rates at which sea turtles shed their scutes and so I am hesitant to make any predictions about how this affects retention rates of epibionts.

• “L255: it would be nice with a reference here and some elaboration of how turtle species differ in these properties. It appears a bit vague.” We do not know of any suitable references for this section so we have instead listed how this could be a productive avenue for future research.

• “L266-267: This needs a reference or proper photo documentation. Is it the cyprid larva that digs into the tissue or is it the adults? Is the burrowing similar to the balanomorphan barnacle Xenobalanus globicipitis that settles and burrows into the integument of whales? If no, what is the difference? Have the two modes of “digging” been observed?” Instead of including extra photos, we have instead cited a suitable study that includes the photos you metion.

• “L306: This is plausible, but as the authors note, barnacles were scraped off with instruments. Barnacles generally attach firmly. I do not think it is likely that they are being scraped off, although some might. Would selection not impede settlement in those areas if it affected the survival of the species (i.e., they fall off and die)? Just a thought. I can buy it if the authors do not wish to go into detail.” Barnacles on the carapace would only be scraped off during reproductive events, which can be many years apart, so we would not think that this would form a strong evolutionary pressure to not settle on the carapace. Also, even though we used instruments to scrape of many of the barnacles, many can be removed by hand so I do think that it could be possible for them to scraped off during reproduction.

We are highly thankful of Reviewer 2’s attention to detail in their comments. They have helped to “fine-tune” much of the wording and scientific arguments in this manuscript.

Thank you for considering our revised and improved manuscript for publication. We are excited to receive your feedback on the latest edition of this manuscript and we look forward to hearing from you soon.

Sincerely,

Nathan J. Robinson

---

## [Decision Letter · Decision Letter 1]

23 Aug 2019

Spatial distribution of epibionts on olive ridley sea turtles at Playa Ostional, Costa Rica

PONE-D-19-15745R1

Dear Dr. Robinson,

We are pleased to inform you that your manuscript has been judged scientifically suitable for publication and will be formally accepted for publication once it complies with all outstanding technical requirements.

With kind regards,

Benny K.K. Chan, Ph.D

Academic Editor

PLOS ONE

Additional Editor Comments (optional):

Reviewers' comments:

Reviewer's Responses to Questions

**Comments to the Author**

1. If the authors have adequately addressed your comments raised in a previous round of review and you feel that this manuscript is now acceptable for publication, you may indicate that here to bypass the “Comments to the Author” section, enter your conflict of interest statement in the “Confidential to Editor” section, and submit your "Accept" recommendation.

Reviewer #1: All comments have been addressed

Reviewer #2: All comments have been addressed

2. Is the manuscript technically sound, and do the data support the conclusions?

Reviewer #1: Yes

Reviewer #2: Yes

3. Has the statistical analysis been performed appropriately and rigorously? 

Reviewer #1: Yes

Reviewer #2: Yes

4. Have the authors made all data underlying the findings in their manuscript fully available?

Reviewer #1: Yes

Reviewer #2: Yes

5. Is the manuscript presented in an intelligible fashion and written in standard English?

Reviewer #1: Yes

Reviewer #2: Yes

6. Review Comments to the Author

Reviewer #1: Authors have carefully addressed all the comments and improved the story flow.

I recommend acceptance.

265 help therefore help

338 rstudies

Reviewer #2: The authors have addressed all comments sufficiently and I recommend to accept the manuscript for publication in PlosOne.

7. PLOS authors have the option to publish the peer review history of their article (what does this mean?). If published, this will include your full peer review and any attached files.

Reviewer #1: No

Reviewer #2: No

---

## [Editor Report · Acceptance letter]

27 Aug 2019

PONE-D-19-15745R1 

Spatial distribution of epibionts on olive ridley sea turtles at Playa Ostional, Costa Rica 

Dear Dr. Robinson:

I am pleased to inform you that your manuscript has been deemed suitable for publication in PLOS ONE. Congratulations! Your manuscript is now with our production department. 

With kind regards,

on behalf of

Dr. Benny K.K. Chan 

Academic Editor

PLOS ONE